# Spatiotemporal dynamics of grassland aboveground biomass in northern China and the alpine region: Impacts of climate change and human activities

**Xinyun Wang** [1] *, **Ji Zhu**[2], **Peipei Pan** [3]

**1** School of Ecology and Environmental Sciences, Ningxia University, Yinchuan, Ningxia, China, **2** College of Land Science and Spatial Planning, Hebei Geo University, Shijiazhuang, Hebei, China, **3** School of Geographical Science, Hebei Normal University, Shijiazhuang, Hebei, China

* wxy_whu@163.com

**Data Availability Statement:** All relevant data are within the paper and its Supporting information files. The data can also be accessed at https://zenodo.org/records/14164654.

## Abstract

Grassland plays a crucial role in the global cycles of matter, energy, water and, climate regulation. Biomass serves as one of the fundamental indicators for evaluating the ecological status of grassland. This study utilized the Carnegie-Ames-Stanford Approach (CASA) model to estimate Net Primary Productivity (NPP) from meteorological data and the Global Inventory Monitoring and Modeling System (GIMMS) Normalized Difference Vegetation Index (NDVI) remote sensing data for northern China's temperate and alpine grasslands from 1981 to 2015. NPP was subsequently converted into aboveground biomass (AGB). The dynamic changes in grassland AGB were analyzed, and the influence of climate change was examined. The results indicate strong agreement between AGB estimations from the CASA model and Gill method based on field-measured AGB, confirming the model's reliability for these regions. The dynamic changes in AGB exhibited a significant increasing trend of 1.31 g/m². Grazing intensity (GI), soil moisture, and mean annual precipitation are identified as key factors influencing changes in grassland AGB. Our findings indicate that precipitation and soil moisture are the primary drivers of AGB accumulation during the growing season (spring, summer, and autumn), while temperature plays a critical role in supporting biomass accumulation during winter. Higher temperatures in winter contributes to increased AGB in the following spring, particularly in desert steppe and alpine meadow ecosystems. These insights highlight the complex interaction between climate factors and human activities in shaping grassland productivity across different seasons.

## 1. Introduction

Climate change significantly influences the service functions and biodiversity of terrestrial ecosystems [1], which, in turn, affects human survival and development. This issue has attracted extensive attention in recent years. As the core component of the biosphere, changes in

**Funding:** This research was supported by the Natural Science Foundation of Ningxia Province under Project NO. 2022AAC03097.

**Competing interests:** The authors have declared that no competing interests exist.

vegetation growth reflect the ways which climate influences our planet and play a crucial feed-back role in the Earth system through the biogeophysical processes [2]. The effects of climate change and human activities on vegetation phenology, the distribution patterns of terrestrial ecosystems, ecosystem processes, structures, and functions, as well as the response and adaptation characteristics of ecosystems, are among the most concerning topics in current and future global change research [3, 4].

Grasslands, which cover more than 30% of the Earth's terrestrial surface [5, 6], are essential for global carbon cycling and climate regulation [7]. In China, grasslands cover 40% of the terrestrial surface [8] and play a vital role in energy exchange, water cycling, and biogeochemical processes [9], as well as in providing resources for animal husbandry, food products, and fuel. The interaction between climate change and grassland ecosystems, significantly influenced by climate warming and human activities, is a key issue in global changes studies [6]. The interaction affects grassland phenology, spatial distribution, and productivity [3, 6, 10].

Aboveground biomass (AGB), defined as the dry weight of the aboveground mass of grass per unit area [11], is a fundamental indicator of a grassland's ecological status and is crucial for assessing productivity and sustainability [12, 13]. Accurately estimating grassland AGB is essential for understanding grassland dynamics and managing grassland resources effectively [14]. AGB can be estimated through various methods, including field surveys, statistical regression models, the combination of optical remote sensing data with radiative transfer models (RTM), and light use efficiency (LUE) models [15, 16]. While field surveys provide accurate data, they are labor-intensive and spatially limited [17]. Remote sensing techniques offer broad coverage and are commonly used to estimate ecological parameters using vegetation indices (VIs) such as the normalized difference vegetation index (NDVI), the ratio vegetation index (RVI), the enhanced vegetation index (EVI), and the modified soil adjusted vegetation index (MSAVI) [15, 18]. Studies have shown that these vegetation indices or spectral reflectance data strongly correlate with biomass [19]. Some researchers have combined multiple variables, including VIs and environmental factors, to estimate grassland AGB [13, 17]. Zeng et al. [13] employed four machine-learning models to estimate grassland AGB and analyzed its response to climatic factors using field observations, remote sensing, meteorological, and topographical data. Their results indicated that the random forest model outperformed the other models. Similarly, Quan et al. [16] estimated grassland AGB in the Qinghai Lake watershed using PRO-SAILH (PROSPECT+SAILH) model and applied partial least squares regression (PLSR) and artificial neural network (ANN). Their results demonstrated that the RTM-based method achieved higher accuracy than the regression and ANN methods.

LUE models, such as the Carnegie-Ames-Stanford Approach (CASA) and the Global Production Efficiency Model (GLO-PEM), integrate remote sensing and meteorological data to simulate aboveground net primary productivity (ANPP) at various scales. The CASA model, in particular, estimates ANPP using the fraction of photosynthetically active radiation (FPAR) and the light energy use rate, and it has been widely applied to estimate vegetation ANPP at regional and global scales. The model has been validated in various settings. Liu et al. [20] investigated the spatiotemporal dynamics of grassland ANPP in China based on the CASA model and meteorological data, revealing that the alpine and sub-alpine meadow exhibited the largest annual mean NPP, while desert grassland showed the smallest. Fang et al. [21] used nar-row-band red-edge information to enhance the accuracy of the AGB mapping for crops based on the CASA model, showing that integrating narrow-band red-edge data improved the accuracy of FPAR and AGB estimates.

Climate change will directly and indirectly influence vegetation productivity [22]. The dynamics of grassland AGB are influenced by various environmental factors and climatic factors. Many studies have analyzed the contribution of different climatic variables to the

spatiotemporal dynamics of grassland AGB [23]. While traditional research often focuses on changes in precipitation and temperature as the primary drivers of grassland biomass dynamics [5, 24], recent studies have increasingly showed the crucial role of seasonal and intra-annual variations in climate patterns in regulating biomass dynamics [24]. These variations can have both positive and negative impacts on AGB [25]. However, they are often overlooked in evaluations of grassland AGB, particularly at large spatial and temporal scales. Wang et al. [24] demonstrated that various environmental drivers, including soil properties, available water capacity, and nutrient levels, and human-induced factors like grazing intensity (GI), together influence changes in grassland AGB. Soil type and nutrient availability directly affect plant growth and biomass accumulation. For example, soils with higher nutrient levels promote more vigorous vegetation growth, leading to increased AGB [26, 27]. In contrast, insufficient soil moisture can limit biomass accumulation, particularly in arid and semi-arid regions. Furthermore, grazing intensity can alter plant community structure and composition, ultimately affecting AGB [28, 29]. Thus, a comprehensive understanding of both climatic and human activity drivers is essential for accurately evaluating grassland AGB dynamics and its response to climatic changes.

Northern China's temperate and alpine grasslands comprise various grassland types and soil properties [30]. These regions are particularly vulnerable to fluctuations in climate, such as temperature and precipitation, as well as environmental conditions like soil moisture and nutrient availability, alongside human activities like grazing intensity [31]. Therefore, considering the impacts of climatic, environmental, and human factors on grassland AGB is crucial for studying the dynamics of carbon cycling in these ecosystems [32].

This study utilized the CASA model to estimate the grassland AGB in the northern China's temperate and alpine regions and examined the effects of climate change and human activities on grassland dynamics. The objectives are twofold: (1) to estimate grassland AGB using the CASA model; and (2) to analyze the spatiotemporal dynamics of grassland AGB in response to climate change. This comprehensive approach aims to provide a deeper understanding of how environmental and human activities drivers shape grassland dynamics and support the sustainable management of these critical ecosystems.

## 2. Materials and methods

### 2.1. Study area

The study was conducted across six provinces: Inner Mongolia, Ningxia, Gansu, Qinghai, Tibet, and Sinkiang (Fig 1). The region spans longitudes of 73˚27' E to 126˚E and latitudes of 26˚52' N to 53˚22' N. It experiences a range of climatic conditions, with mean annual temperature (MAT) varying from -3.1 to 8.9˚C and mean annual precipitation (MAP) from 62.7 mm to 694 mm. The region is primarily characterized as arid and semi-arid. The grasslands in the study area are classified into nine types based on the MODIS land use and land cover data [33]: alpine meadow, alpine steppe, temperate meadow steppe, temperate steppe, montane meadow, warm-temperate tussock and shrub-tussock, tropical tussock and shrub-tussock, as well as temperate desert steppe [34].

### 2.2. Data

**2.2.1 Grassland type data.** The MODIS MCD12C1 dataset, with a spatial resolution of 0.05˚, was used to classify land cover/vegetation types in the study areas [33]. The dataset was processed using MODIS Reprojection Tool (MRT). Grasslands are classified into nine types based on this dataset: alpine meadow, alpine steppe, temperate meadow steppe, temperate

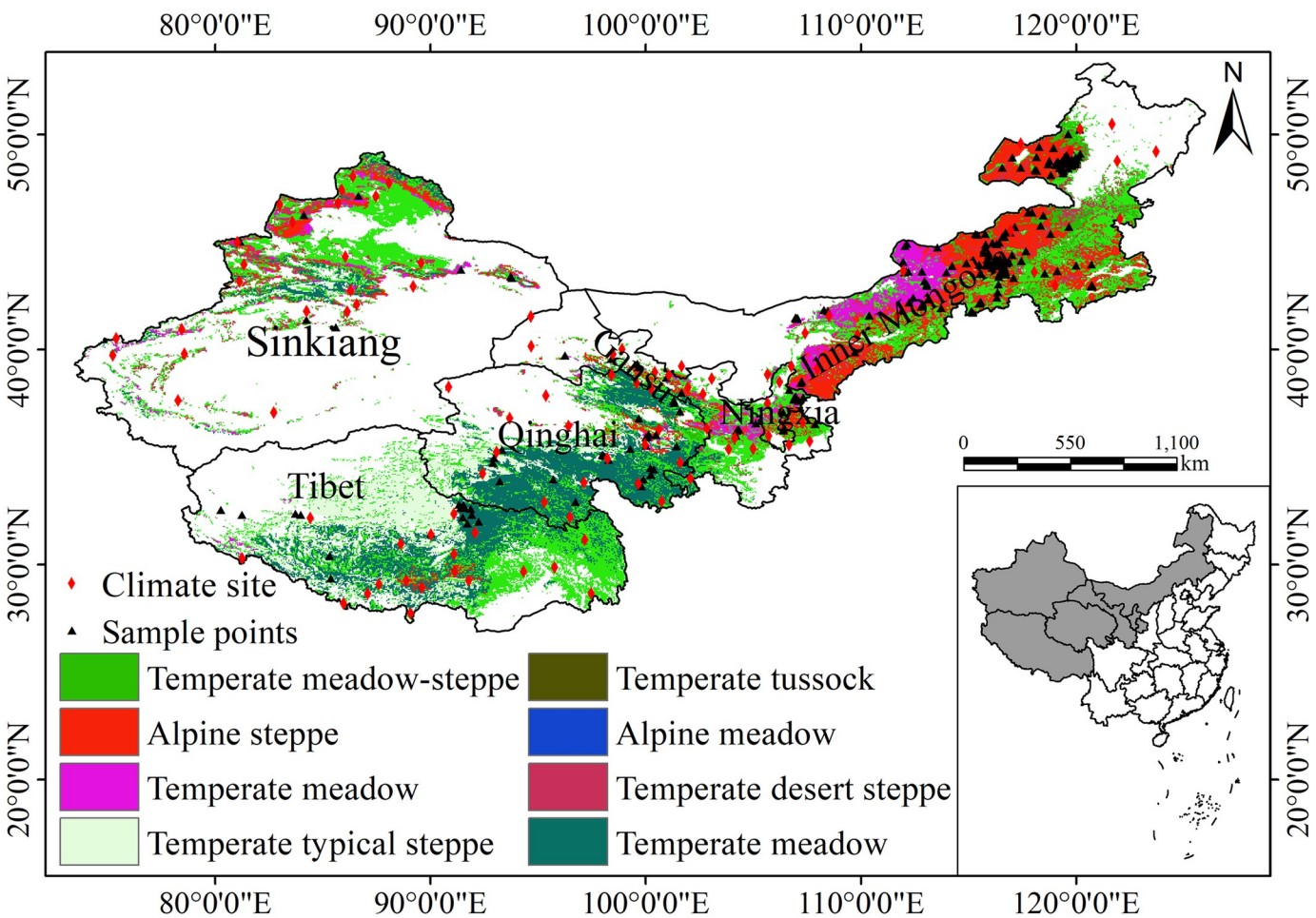

**Fig 1. Spatial distribution of the study area and grassland types in China.** The blank areas mark the locations of the study areas on the map of China. Black stars indicate the locations of climate stations, and red triangles represent ground measurement points. Source: Land use and cover data is from the Land Processes Distributed Active Archive Center (LP DAAC, https://lpdaac.usgs.gov). The study areas and the map of China are from National Geomatics Center of China (https://www.ngcc.cn), and the meteorological data are from the China Meteorological Administration (CMA, http://www.nmic.cn).

steppe, montane meadow, warm-temperate tussock and shrub-tussock, tropical tussock and shrub-tussock, and temperate desert steppe [34].

**2.2.2. GIMMS NDVI data.** The Global Inventory Monitoring and Modeling System (GIMMS) Normalized Difference Vegetation Index (NDVI) 3g v1 dataset was used to assess grassland dynamics. This dataset, derived from the third-generation (3g) Advanced Very High Resolution Radiometer (AVHRR) sensor onboard the National Oceanic and Atmospheric Administration (NOAA) satellites, spans the period from 1981 to 2015 at a resolution of 0.083˚ [35]. Preprocessing included radiation calibration, geometric correction, and cloud removal [36]. Missing values in the NDVI data were addressed through linear interpolation. Gaps in the GIMMS NDVI3g time series were smoothed using Savitzky-Golay (SG) filters [37, 38], and outlier values were detected and removed using the interquartile range (IQR) method [39]. Monthly NDVI values were derived using the Maximum Value Composite (MVC) method [40], and the data were resampled to a resolution of 0.05˚.

**2.2.3. Meteorological data.** Daily meteorological data from 1981 to 2015 were sourced from the China Meteorological Administration (CMA). These records included air

temperature (T), precipitation (PPT), solar radiation (SR), and relative humidity (RH) collected from 99 meteorological stations. Missing values in the meteorological data were handled through linear interpolation, while outliers were detected and removed using the IQR method to ensure data integrity. The data were spatially interpolated to a grid with a resolution of 0.05˚ using the Anusplin software (https://fennerschool.anu.edu.au/). Vapor pressure deficit (VPD) was calculated using the Penman-Monteith and Priestley-Taylor formulas [41, 42].

**2.2.4. Soil moisture.** The ERA5 dataset, developed by the European Centre for Medium-Range Weather Forecasts (ECMWF), is the fifth generation of global atmospheric reanalysis data [43]. It provides time-series meteorological reanalysis data from 1950 to present. This study utilized the ERA5 Land hourly dataset, which includes soil moisture (SM) data from 1981 to 2015 with a resolution of 0.1˚. To comprehensively evaluate the impacts of soil moisture on AGB across the 0–28 cm depth, soil moisture values from the first (0–7 cm) and second (7–28cm) layers were combined, using the thickness of each layer as a weight factor [44, 45]. All SM data were then resampled to a spatial resolution of 0.05˚.

## 2.3. Estimation and validation of grassland AGB using the CASA model

The CASA model was employed to estimate vegetation Net Primary Productivity (NPP) based on absorbed photosynthetically active radiation (APAR) and light use efficiency (LUE). Monthly NPP for each pixel is expressed as follows [46]:

$$NPP\,(x, t) = APAR\,(x, t) \text{ x } LUE\,(x, t), \qquad (1)$$

Where APAR(x, t) represents the solar radiation absorbed by vegetation (MJ/m$^2$), and LUE(x, t) denotes the light use efficiency (g C/MJ).

NPP was converted to AGB using a model developed by Gill et al. [47], which divides NPP into aboveground (ANPP) and belowground (BNPP) components. AGB is calculated as follows:

$$AGB = \frac{ANPP}{R/S} \times (C_a + C_b), \qquad (2)$$

where R/S represent the root-to-short ratios, and $C_a$, and $C_b$ denote the carbon content of aboveground and belowground biomass, respectively, specific to each grassland type (listed in Table 1).

Field-measured biomass data from Hu et al. [48] and Ma et al. [49] were utilized to validate model-derived AGB estimates from 370 sample plots in 2001 and 2010 (Fig 1). The carbon density in AGB was estimated using a coefficient of 0.45.

**Table 1. Parameters for R/S, $C_a$, and $C_b$ across different grassland types.**

| Grassland types | R/S | Ca | Cb |
|---|---|---|---|
| Temperate meadow steppe | 5.3 | 0.43 | 0.37 |
| Temperate typical steppe | 5.3 | 0.43 | 0.37 |
| Temperate desert steppe | 6.7 | 0.43 | 0.37 |
| Alpine steppe | 5.2 | 0.43 | 0.37 |
| Tropical tussock | 1.6 | 0.43 | 0.37 |
| Temperate meadow | 6.3 | 0.43 | 0.37 |
| Alpine meadow | 6.8 | 0.43 | 0.37 |

## 2.4. Grazing intensity estimation

Grazing intensity (GI) was assessed using the formula derived from Zhai et al. [50]. GI reflects the balance between livestock feed intake and the available grassland biomass. GI is calculated as follows:

$$GI = \frac{ANNP_t - AGB_t}{P \times T},$$ (3)

where ANPP$_t$ and AGB$_t$ represent the aboveground net primary productivity and biomass at time t, respectively. T represents the number of grazing days, and P denotes the daily feed intake of livestock.

## 2.5. Data analysis methods

We used Random Forest (RF) to analyze the relationships between grassland AGB and various driving variables, including temperature (T), precipitation (PPT), vapor pressure deficit (VPD), solar radiation (SR), relative humidity (RH), soil moisture (SM), and grazing intensity (GI). The RF model, implemented in R using the "ranger" package [51], was configured with 1000 trees.

To assess vegetation dynamics through time-series data, we used Theil-Sen Median trend analysis and Mann-Kendall test to assess temporal trends in grassland AGB from 1981 to 2015 [52–54]. The coefficient of variation (CV) is used to assess the consistency of grassland productivity in response to environmental changes and management practices [55].

Pearson correlation coefficients were calculated to examine relationships between grassland AGB and the driving variables across different pixels and years [20, 56]. Correlations were categorized based on both the correlation coefficient and the results of t-test as follows: (1) Significant negative (SN, $r < 0$, $p < 0.05$); (2) Non-significant negative (NSN, $r < 0$, $p > 0.05$); (3) Non-significant positive (NSP, $r > 0$, $p > 0.05$); (4) Significant positive (SP, $r > 0$, $p < 0.05$).

## 3. Results

### 3.1. Validation of grassland AGB

The estimation of grassland AGB was conducted using the CASA model combined with the Gill method. The validation of the model was performed by comparing model-estimated AGB with field-measured data. For this purpose, a dataset comprising 370 field measurements from the years 2001 and 2010 was used. The comparative results are depicted in Fig 2.

As illustrated in Fig 2a, the model's predictions in 2001 showed a coefficient of determination ($R^2$) of 0.62, indicating moderate accuracy with a p-value less than 0.05. The Root Mean Square Error (RMSE) was 43.50 g/m$^2$. It was observed that the estimated AGB at certain points was higher than the field measurements, leading to an overestimation of field-measured AGB. In contrast, Fig 2b represents the relationship between the model-predicted and field-measured AGB for 2010, showing a strong linear relationship. The solid line represents the regression fit, while the dotted line represents the ideal 1:1 line correspondence between predicted and measured values. For 2010, the model achieved a high accuracy with an $R^2$ of 0.79 and an RMSE of 27.46 g/m$^2$. Most data points closely followed the ideal 1:1 line, validating the model's robustness in predicting grassland AGB.

### 3.2. Spatial distributions of annual and seasonal changes in grassland AGB

**3.2.1 Annual distribution of grassland AGB.** Significant spatial variations in AGB were observed across the grasslands of northern China and alpine regions. As illustrated in Fig 3,

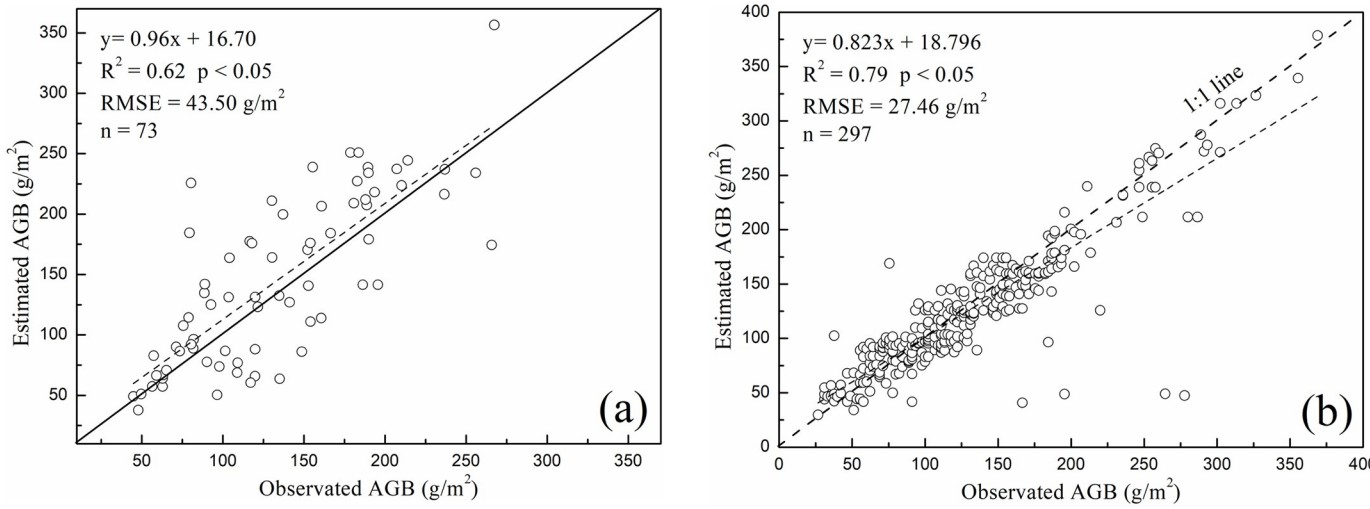

**Fig 2. Comparison of field-measured and estimated AGB (g/m²) for (a) 2001 and (b) 2010.**

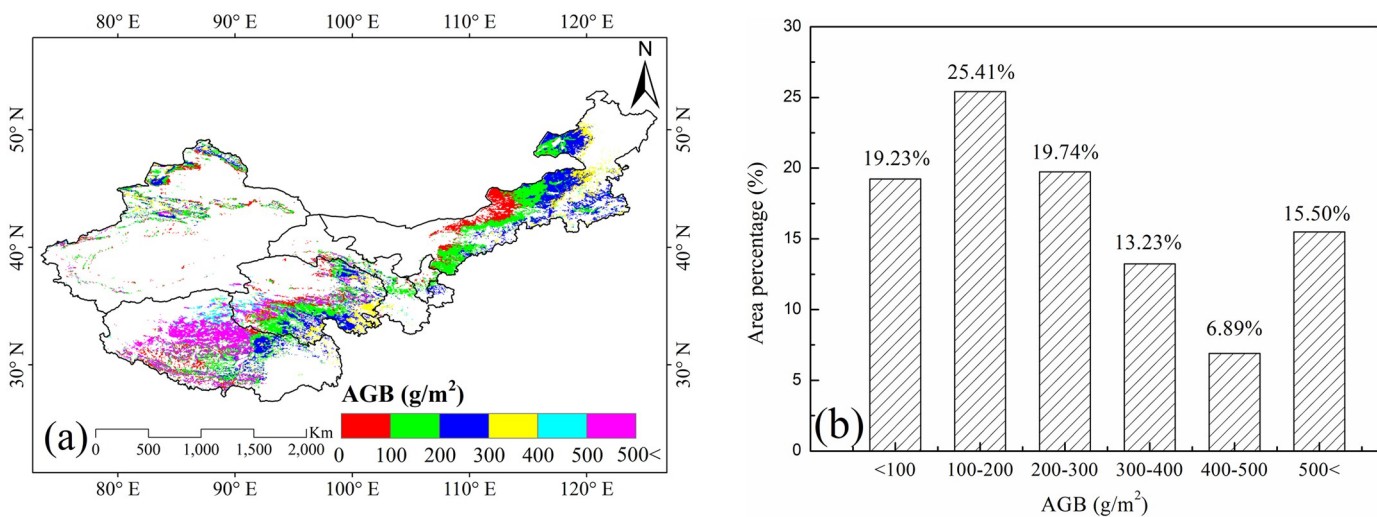

**Fig 3.** (a) Spatial distribution of grassland AGB (g/m²); (b) Percentage of area covered by different AGB ranges. Source: Land use and cover data is from the Land Processes Distributed Active Archive Center (LP DAAC, https://lpdaac.usgs.gov). The study areas are from National Geomatics Center of China (https://www.ngcc.cn).

areas with AGB values below 100 g/m² were predominantly located in Ningxia, the central Inner Mongolia, Qinghai, and much of Sinkiang, accounting for about 19.23% of the total grassland area. AGB values ranging from 100 to 300 g/m² were most common in the majority of Inner Mongolia, Gansu, southern Qinghai, and eastern Tibet, covering approximately 45.15% of the total grassland area. Regions with AGB values ranging from 300 to 400 g/m² were primarily found in eastern Inner Mongolia, eastern Qinghai, and northern Tibet, making up about 13.23% of the grassland area. The highest AGB values, exceeding 500 g/m², were concentrated in central Tibet, and central and western Qinghai, covering about 15.50% of the total grassland area. Overall, grasslands in the study area had an average AGB of

429.01±56.24 g/m$^2$, spanning the largest area of approximately 75,080 km$^2$. Conversely, the temperate desert steppe showed the lowest AGB, with an average of 67.34±10.50 g/m$^2$ (S1 Table).

**3.2.2. Seasonal distribution of grassland AGB.**   The seasonal distribution of grassland AGB was also investigated by dividing the year into four seasons: spring (March to May), summer (June to August), autumn (September to November), and winter (December to February). The results, shown in Fig 4, reveal that AGB exhibits distinct seasonal fluctuations. AGB reaches its peak in summer, with values approximately 120 g/m$^2$. In contrast, AGB remains relatively low, primarily between 0 and 20 g/m$^2$ during spring, autumn, and winter. The highest values are observed during the growing season (spring to summer), followed by a gradual decrease in autumn and winter. These seasonal changes highlight the importance of considering seasonal dynamics in AGB assessments, which may be influenced by variations in precipitation and temperature throughout the year [22].

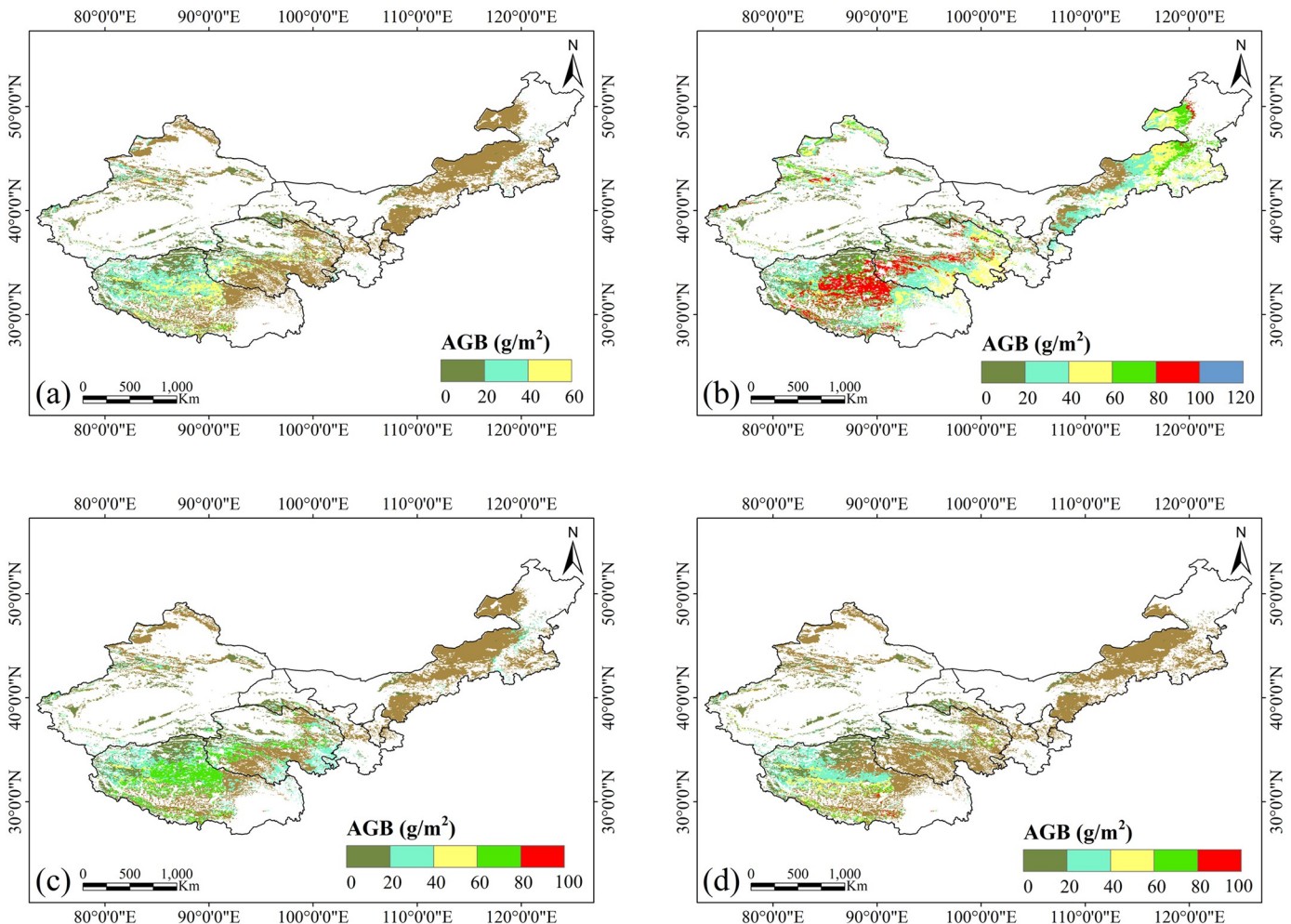

**Fig 4. Seasonal distributions of grassland AGB: (a) Spring, (b) Summer, (c) Autumn, and (d) Winter.** Source: Land use and cover data is from the Land Processes Distributed Active Archive Center (LP DAAC, https://lpdaac.usgs.gov). The study areas are from National Geomatics Center of China (https://www. ngcc.cn).

## 3.3. The temporal and spatial change trends of grassland AGB from 1981 to 2015

**3.3.1. Temporal change trends of grassland AGB.** Fig 5 illustrates the inter-annual variability of grassland AGB from 1981 to 2015, showing notable fluctuations over this period. The highest AGB value was recorded in 2015, at 415.64 g/m$^2$, while the lowest occurred in 1997, at 222.01 g/m$^2$. Overall, the trend analysis indicates a significant increase in AGB with an average annual slope of 1.31 g/m$^2$ (r = 0.36, p < 0.05). The temporal dynamics of grassland AGB can be divided into two distinct phases: a decline from 1981 to 2005, with a slope of -1.18 g/m$^2$, followed by a shape increase of 15.01 g/m$^2$ per year from 2005 to 2015. Throughout most of the study period, AGB values were below the mean of 272.72 g/m$^2$ (Fig 5a).

Fig 5b and S2 Table detail the AGB inter-annual trends for the seven main grassland types. The AGB showed increasing trends in temperature typical steppe, temperate desert steppe, alpine steppe, temperate meadow, and alpine meadow from 1981 to 2015. Notably, the alpine steppe exhibited the most significant increase in AGB, with a growth rate of 3.313 g/m$^2$, while the temperate typical steppe experienced the smallest increase at only 0.061 g/m$^2$. Conversely, the AGB of temperate meadow steppe and tropical tussock showed decreasing trends with the steepest decline occurring in the tropical tussock (-2.965 g/m$^2$) and a milder decrease in the temperate meadow steppe (-0.262 g/m$^2$).

**3.3.2. Spatial change trends and stability analysis of grassland AGB.** The spatial dynamics of grassland AGB from 1981 to 2015 showed considerable regional variation (Fig 6a). Notably, 21.61% of the total grassland area, predominantly located in northwest Tibet and the western Qinghai, exhibited significant improvement in AGB (S3 Table). A slight improvement was observed in 18.99% of the total area, primarily distributed across central Tibet, Qinghai, and Sinkiang. Regions with stable AGB levels constituted 19.36% of the grassland area, dispersed across the six provinces. Conversely, slight degradation in AGB affected 35.54% of the area, particularly in central Inner Mongolia, southern Tibet and Qinghai, and Sinkiang. Significant degradation was observed in 5.49% of the grassland area, mainly in central and eastern Inner Mongolia, as well as in Qinghai and Sinkiang.

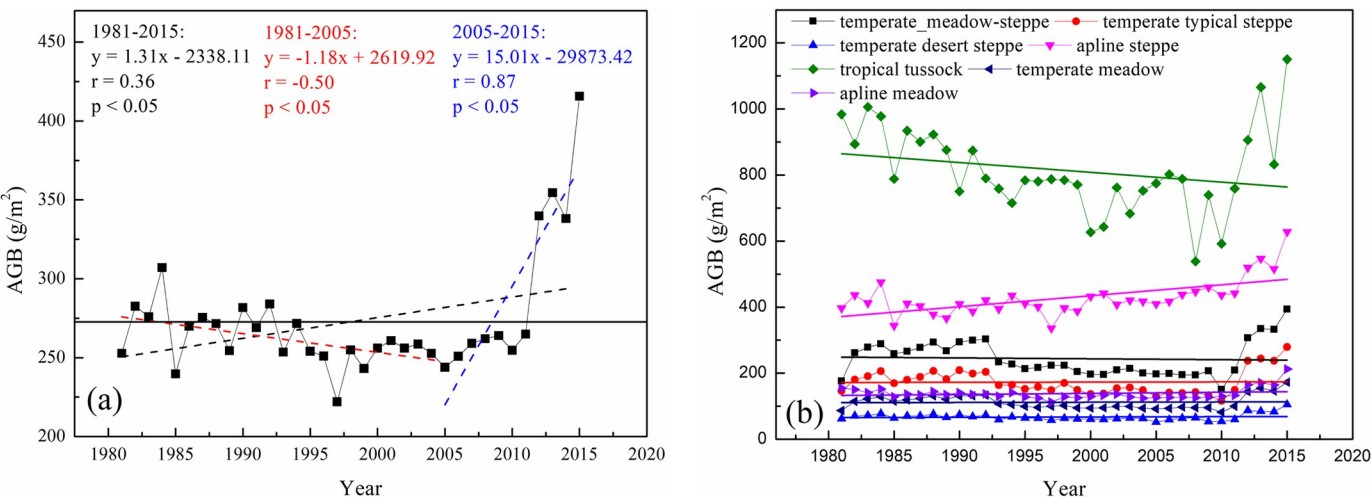

**Fig 5. Inter-annual variation in AGB for (a) all grassland types and (b) seven main grassland types.** Black, red and blue dotted lines represent changes in grassland AGB over the periods 1981–2015, 1981–2005, and 2005–2015, respectively. The black solid line indicates the mean AGB of 272.72 g/m$^2$ for all grassland types.

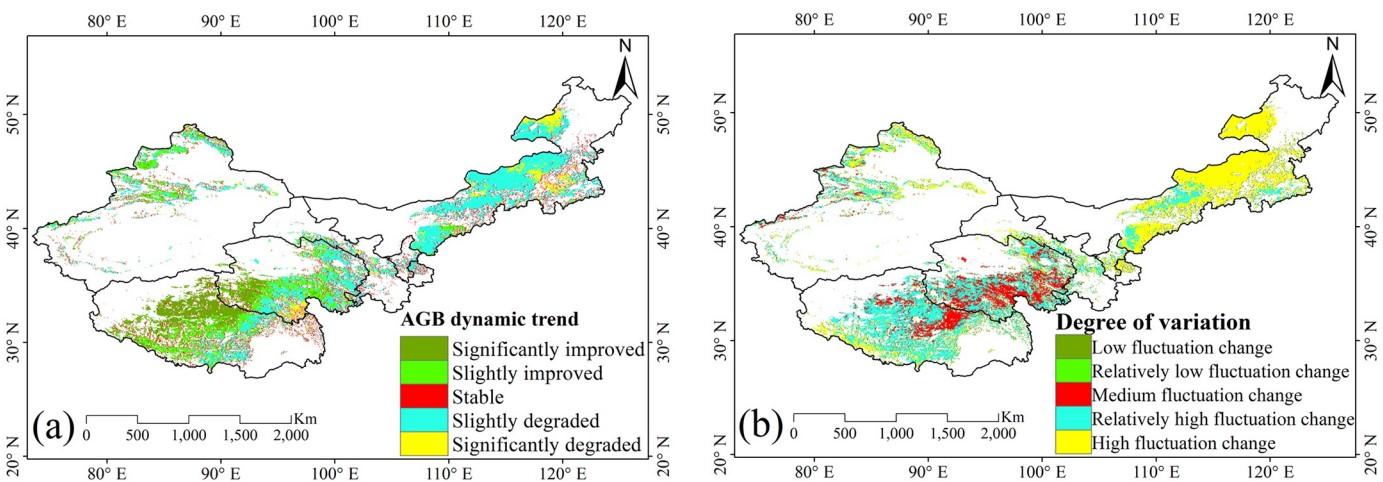

**Fig 6. (a) Spatial trends and (b) coefficient of variation of grassland AGB from 1981 to 2015.** These trends highlight the temporal and spatial variability in grassland productivity over the study period. Source: Land use and cover data is from the Land Processes Distributed Active Archive Center (LP DAAC, https://lpdaac.usgs.gov). The study areas are from National Geomatics Center of China (https://www.ngcc.cn).

The coefficient of variation (CV) showed the degree of fluctuation in grassland AGB, categorized into five levels: low, relatively low, medium, relatively high, and high fluctuation. The spatial distribution of the AGB's CV from 1981 to 2015 showed significant variation (Fig 6b). Grasslands with low to relatively low fluctuation levels, representing 19.20% of the total grassland area (S4 Table), were distributed across the six provinces, which suggests stable grassland growth. Medium fluctuation levels were mainly observed in Tibet and Qinghai, covering 13.07% of the total grasslands area. Areas with relatively high and high fluctuation, accounted for 34.04% and 33.63% of the grassland area, respectively, were primarily located in Tibet, Inner Mongolia, Qinghai, and Sinkiang, indicating more distinct changes in grassland AGB.

## 3.4. Impacts of driving factors on grassland AGB

The relationships between grassland AGB and various driving factors, including mean annual temperature (T), precipitation (PPT), vapor pressure deficit (VPD), solar radiation (SR), relative humidity (RH), soil moisture (SM), and grazing intensity (GI), are detailed in Fig 7. A significant negative correlation between AGB and mean annual temperature was observed across 51.72% of the total grassland area, while 28.47% showed a non-significant negative relationship (Fig 7a). In contrast, only 19.81% of the area showed a non-significant positive correlation, predominantly in the Qinghai—Tibetan Plateau and Sinkiang. This suggest that temperature plays a critical role in influencing AGB across various regions. Higher temperature can alter the growing season and stimulate photosynthetic rate, ultimately leading to changes in AGB [57]. Specifically, higher temperature may increase respiration rates, potentially offsetting the benefits of increased photosynthesis, particularly in arid and semi-arid regions where water availability is a limiting factor [58].

Regarding precipitation, 10.66% of the grassland area, mainly in central Tibet, showed a non-significant negative correlation with AGB. Conversely, 88.04% of the grassland area showed a non-significant positive correlation, indicating that rainfall generally has a beneficial impact on AGB (Fig 7b). The positive correlation indicates that adequate precipitation plays a crucial role in driving grassland growth. Precipitation affects soil moisture retention and nutrient availability for vegetation, supporting vegetation development [57]. For VPD, 28.69% of

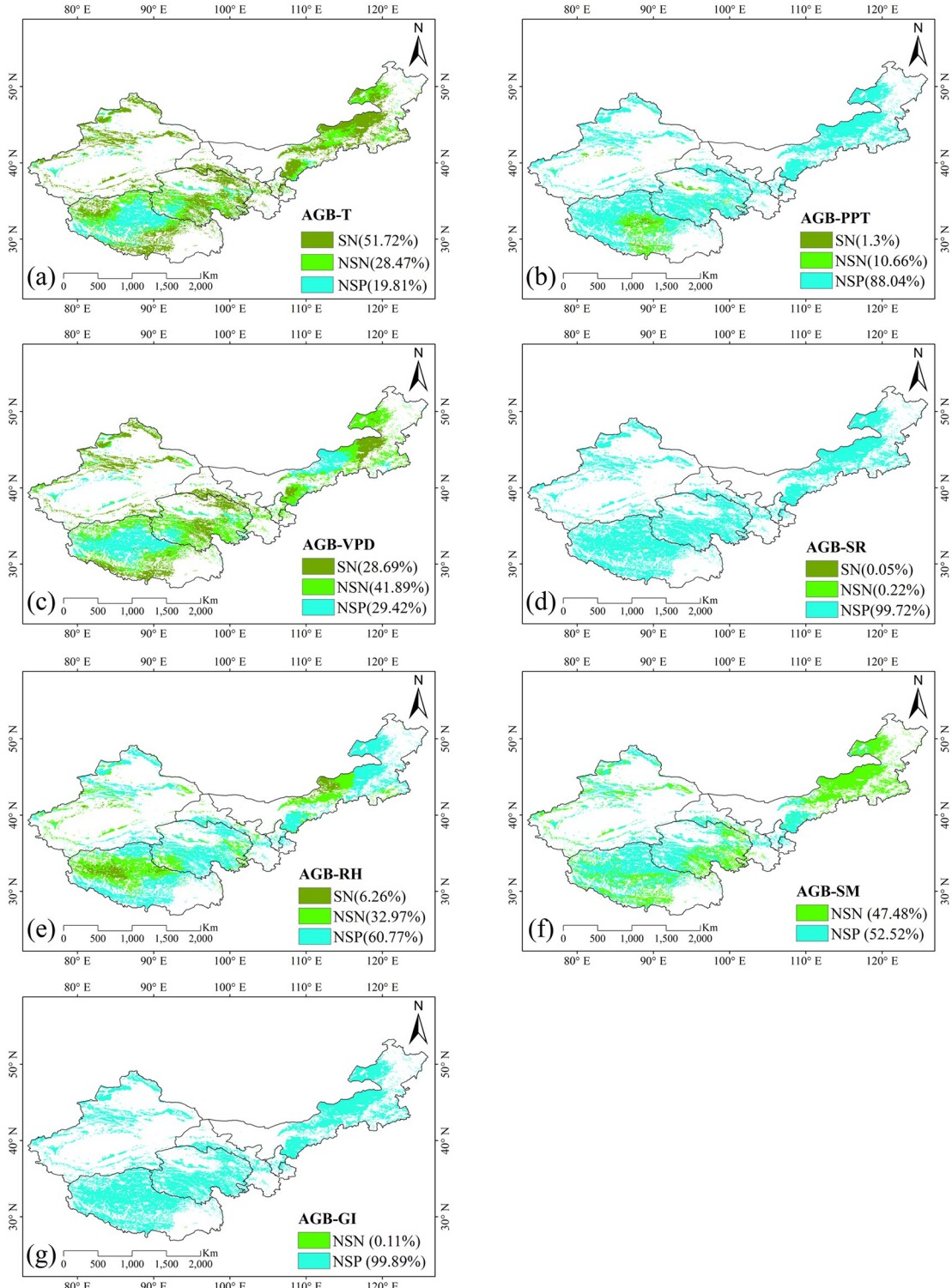

**Fig 7. Correlation between grassland AGB and key factors: (a) mean annual temperature (T), (b) mean annual precipitation (PPT), (c) vapor pressure deficit (VPD), (d) solar radiation (SR), (e) relative humidity (RH), (f) soil moisture (SM), and (g) grazing intensity (GI).** Categories are as follows: SN = Significant negative, NSN = Non-significant negative, NSP = Non-significant positive. Source: Land use and cover data is from the Land Processes Distributed Active Archive Center (LP DAAC, https://lpdaac.usgs.gov). The study areas are from National Geomatics Center of China (https://www.ngcc.cn).

the area showed a significant negative correlation, indicating adverse effects on AGB. The negative correlation suggests that higher VPD values increase evaporative demand, which can cause water stress in plants. Such stress can inhibit photosynthesis and hinder vegetation growth, ultimately resulting in decreased AGB [59]. Conversely, 41.89% and 29.42% of the grassland exhibited a significant and non-significant positive relationship with VPD, respectively, particularly in central parts of Qinghai Tibetan—Plateau and Inner Mongolia (Fig 7c). In these regions, moderate VPD levels may enhance gas exchange and promote plant growth, especially in areas where plants are adapted to such conditions.

Solar radiation showed a largely non-significant positive relationship with grassland AGB, affecting 99.72% of the area, indicating that solar radiation generally supports grassland productivity across most regions (Fig 7d). Solar radiation is crucial for photosynthesis because it directly influences the energy availability for plant growth. Higher levels of solar radiation can enhance the photosynthesis rate, leading to increased biomass accumulation in grassland ecosystems. Furthermore, adequate sunlight improves plant health and resilience, enabling grasses to better utilize nutrients and moisture, all of which contribute to increased AGB [22].

Regarding relative humidity (RH), only 6.26% of the area showed a significant negative correlation, while 10.66% showed a non-significant negative correlation, primarily in central Inner Mongolia and the Qinghai—Tibetan Plateau. However, the majority of the grassland area, accounting for 60.77%, showed a non-significant positive correlation with RH, highlighting its generally positive influence on AGB (Fig 7e). Higher RH levels can reduce transpiration rates, minimizing water loss and promoting better hydration for plants. This increased water availability is essential for supporting photosynthesis and plant growth. Conversely, low RH can lead to increased water loss through evaporation and transpiration, potentially inducing stress in plants and negatively affecting AGB [59, 60]. Soil moisture exhibits a non-significant negative correlation with AGB in 47.48% of the area, while a non-significant positive correlation was observed in 52.52% of the area (Fig 7f). These findings suggest that soil moisture plays a critical role in maintaining soil water availability, which influences vegetation growth and AGB accumulation. Interestingly, although most areas show a non-significant positive correlation between grazing intensity and AGB, covering 99.89% of the area (Fig 7g), this may be due to the role of moderate grazing in enhancing soil moisture retention and supporting root development. In arid and semi-arid regions, grazing can create conditions that facilitate better water availability, ultimately leading to an increase in AGB [61].

Seven driving factors, including temperature (T), precipitation (PPT), vapor pressure deficit (VPD), solar radiation (SR), relative humidity (RH), soil moisture (SM), and human activities such as grazing intensity (GI), were analyzed for their impact on grassland AGB. We employed a Random Forest (RF) model to assess the relative influence of each factor. The variable importance was plotted along the x-axis and y-axis to reflect their significance in altering grassland AGB. A higher importance value indicates a greater impact on AGB, while a lower value indicates a lesser impact.

Fig 8 illustrates the relative importance of these factors in influencing grassland AGB. The results reveal that GI, SM, and PPT are the predominant factors affecting grassland AGB, each contributing over 10% to the variation in AGB. Grazing intensity was the most critical factor, accounting for 49.19% of the variations in AGB, followed by soil moisture, which contributed 17.46%. Precipitation was the third most influential factors, accounting for 11.93%. In contrast, T, VPD, SR, and RH contributed less than 10% to AGB changes, indicating their relatively minor roles in grassland AGB dynamics. The results indicate that driving factors such as grazing intensity, soil moisture, and precipitation play important role in promoting plant growth and biomass accumulation [59]. However, the roles of other factors, such as mean annual

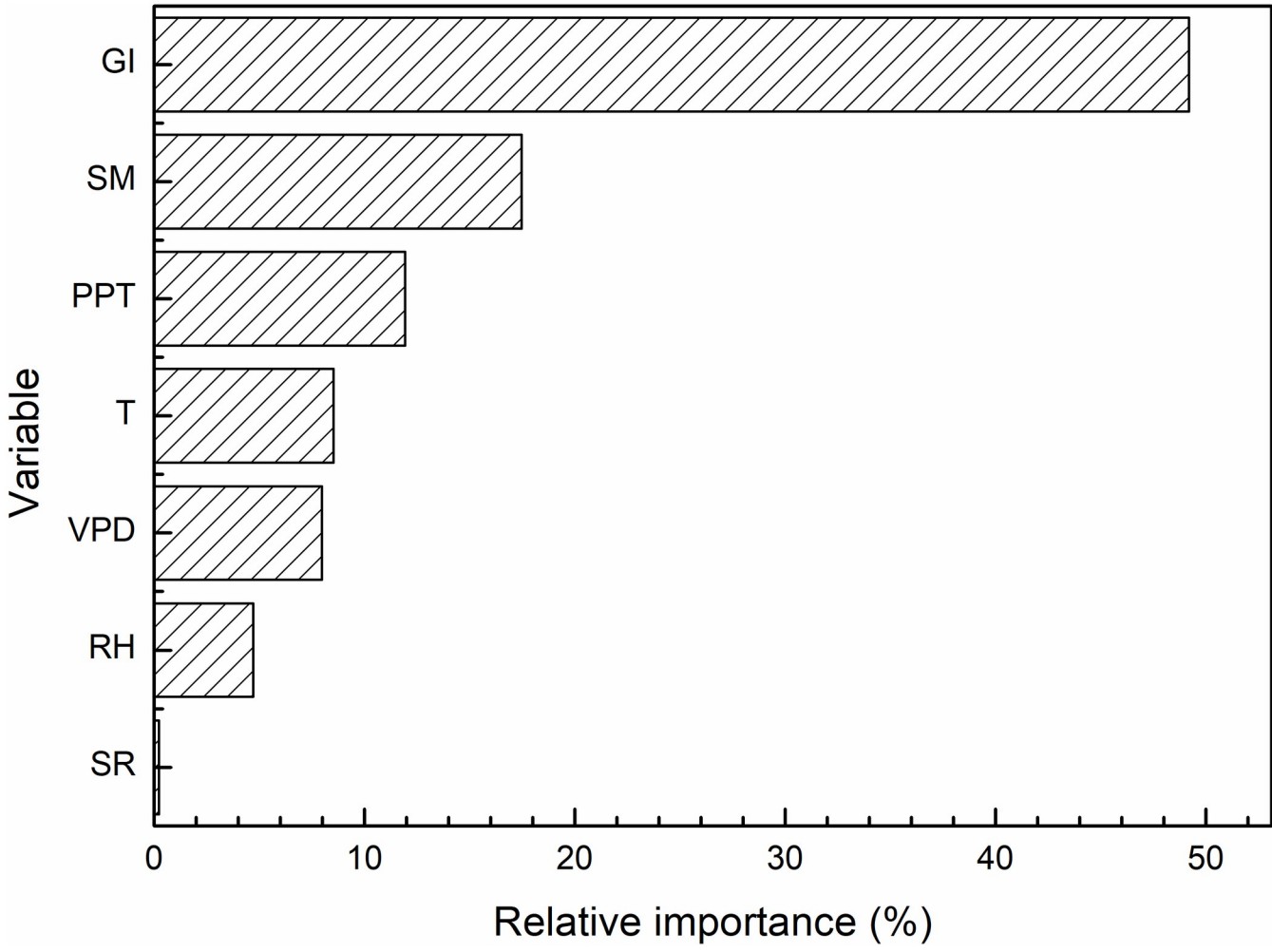

**Fig 8. Relative importance of driving factors on grassland AGB as assessed by Random Forest analysis.**

temperature, vapor pressure deficit (VPD), relative humidity, and solar radiation should not be overlooked in influencing grassland AGB.

## 3.5. Correlation analysis of AGB and driving factors across various grassland types

The correlations (r values) between AGB and various influencing factors across different grassland types were analyzed (Fig 9). Overall, grassland AGB showed negative correlations with annual mean temperature across all grassland types. However, in certain regions, such as typical steppe, desert steppe, temperature meadow, and alpine meadow, AGB showed positive correlations with air temperature (Fig 9a). This suggests that increasing temperature may limit biomass accumulation in many regions. It can also promote vegetation growth in specific areas, particularly in regions where plants have adapted to warmer conditions. Precipitation exhibited a strong positive correlation with AGB across all grassland types, except in temperate and alpine meadows (Fig 9b). This indicates that adequate water is essential for plant growth and biomass accumulation, especially in arid and semi-arid regions [22, 57]. The relationships

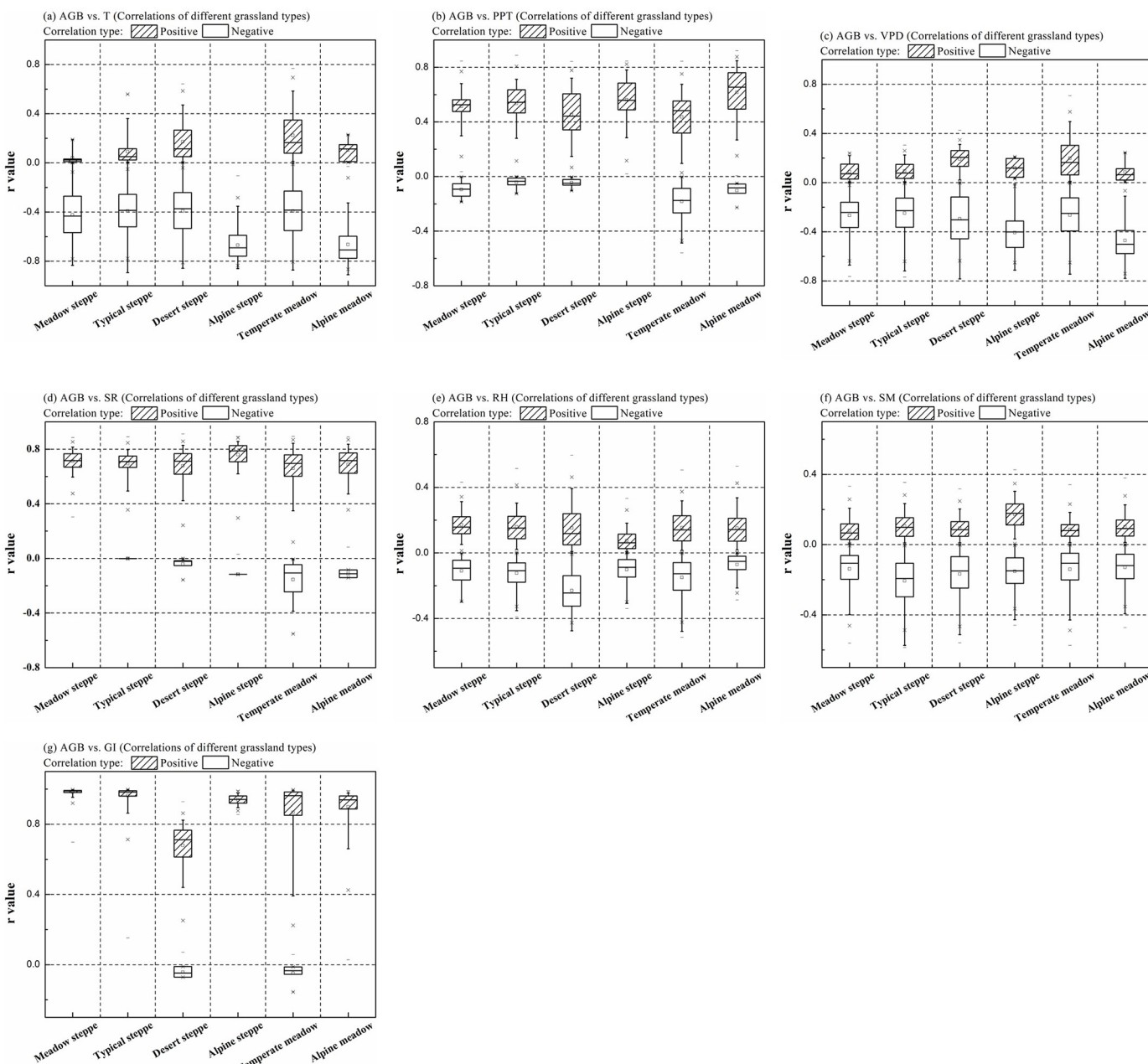

**Fig 9. Corrections (r values) between grassland AGB and driving factors: (a) air temperature (T), (b) precipitation (PPT), (c) vapor pressure deficit (VPD), (d) solar radiation (SR), (e) relative humidity (RH), (f) soil moisture (SM), and (g) grazing intensity (GI) across different grassland types.**

between AGB and other factors such as VPD, RH, and SM showed both positive and negative correlations across different grassland types (Fig 9c, 9e, and 9f). These variations suggest that the impacts of climatic and soil moisture factors on AGB are influenced by the specific physiological traits and water use efficiencies of plant species within each grassland ecosystem. For example, plants in drier environments may maintain AGB despite low soil moisture or higher VPD, while plants in wetter environments may be more sensitive to changes in RH and VPD, leading to more pronounced effects on AGB. These differences suggest the complex

interaction between climate, soil moisture, and grazing intensity in regulating vegetation growth and AGB accumulation across different grassland ecosystems [59].

AGB consistently demonstrates a strong positive correlation with solar radiation across all grassland types (Fig 9d), highlighting its critical role in supporting photosynthesis and enhancing biomass productivity in vegetation. Additionally, AGB exhibited positive correlations with grazing intensity. However, negative relationships were observed in desert steppe and temperate meadows. This suggests that adaptive grazing can enhance soil nutrient availability and promote plant growth and biomass accumulation in some grassland types [62]. Conversely, in desert steppe and temperate meadows, overgrazing may lead to soil degradation, ultimately reducing grassland AGB. Therefore, while moderate grazing can positively contribute to grassland growth, overgrazing can have detrimental effects on productivity [61].

## 4. Discussion

### 4.1. Spatial distribution characteristics of grassland AGB

This study evaluated grassland AGB across northern China and alpine regions from 1981 to 2015 using meteorological and GIMMS NDVI data, employing the CASA model and Gill method. Similar methodologies have been applied in other studies, such as Jiao et al. [63], which generated a grassland AGB dataset for northern China and Qinghai-Tibet Plateau using GIMMS NDVI3g.v1 data, achieving a model accuracy $R^2 = 0.63$. Our findings corroborate these results, demonstrating strong agreement between estimated and field-measured AGB, which supports the effectiveness of the CASA model in these regions.

The spatial distribution of grassland AGB varied significantly across different grassland types in northern China's temperate and alpine regions. The average AGB from 1981 to 2015 was 272.72 g/m$^2$, with approximately half of the total grassland area having values between 100 and 300 g/m$^2$ (Fig 3). Notably, AGB in alpine steppes of Qinghai—Tibetan Plateau was higher compared to temperate grasslands, while the desert grasslands exhibited the lowest AGB. These findings are consistent with those of Liu et al. [20], who observed similar spatiotemporal dynamics in grassland NPP across China.

Grassland AGB also exhibits obvious seasonal fluctuations, driven by changes in temperature, precipitation, and soil moisture. These driving factors play a critical role in shaping the seasonal growth patterns of grassland vegetation across different ecosystems (S5–S7 Tables). Our results indicate that soil moisture and precipitation generally show a positive relationship with AGB accumulation during spring, summer, and autumn, suggesting that these factors support plant growth by providing adequate water for photosynthesis during the growing seasons. In contrast, during winter, the correlation between soil moisture or precipitation and AGB tends to be weak or negative due to the reduced vegetation growth at low temperature, where water availability is less critical.

Temperature exhibits a negative correlation with grassland AGB during the spring, summer, and autumn in most grassland types. This may be due to the fact that higher temperature increase evapotranspiration and water stress during the growth season, ultimately limiting biomass accumulation. However, in desert steppe, temperature shows a positive correlation with AGB during the warmer seasons, which may be due to desert plants' stronger adaptive capacity to higher temperature. In winter, temperature also exhibits a strong positive correlation with grassland AGB, as higher temperatures contribute to the accumulation of AGB in the following spring in these regions. Overall, our findings suggest that precipitation and soil moisture are key drivers of grassland AGB in growing season (spring, summer, and autumn). On the other hand, temperature plays a critical role in supporting biomass accumulation in winter in certain ecosystems, particularly in desert steppe and alpine meadow.

## 4.2. Dynamics of grassland AGB

The study period saw a general increase in the average AGB across northern China and alpine grasslands, which is consistent with existing literature [20]. Notably increases were observed in several grassland types, including temperate typical steppe, temperate desert steppe, alpine steppe, temperate meadow, and alpine meadow. Conversely, temperate meadow steppe showed a decline in AGB. Restoration efforts such as shrub planting, rotational grazing, and fencing to restrict grazing have significantly promoted grassland recovery and biomass increase in these regions since 1990 [64].

Interestingly, our study also revealed significant fluctuations in AGB from 1981 to 2015, with regions in northwest Tibet, west Qinghai, and northwest Sinkiang showing considerable improvement. These areas benefitted from a warming climate trend (S1 Fig), which generally promoted vegetation growth and biomass increase [9]. In contrast, central Inner Mongolia and the southern regions of Tibet and Qinghai experienced slight degradation in AGB due to increased temperatures and reduced precipitation (S2 Fig).

## 4.3. Impacts of climate factors and human activities on grassland dynamics

The dynamics of grassland vegetation are influenced by a complex interplay of climatic, environmental factors and human activities that vary considerably across temporal and spatial scales [20]. Our findings underscore that temperature and precipitation are pivotal in driving AGB changes across northern China's and alpine grasslands, echoing findings from other studies [65, 66]. For example, Liu et al. [20] observed similar patterns in AGB responses to temperature increases. However, our study extends this understanding by focusing on specific grassland types, including alpine meadows and desert steppes, which exhibit varied responses to temperature changes. Additionally, we found that precipitation plays a crucial role in driving changes in AGB, particularly in plain areas and desert grassland [20].

While increasing temperatures may be beneficial in some regions by extending the growing season, they can increase the frequent of meteorological drought and subsequent reductions in AGB in arid and semi-arid areas. Conversely, rising precipitation levels have generally promoted vegetation growth and enhanced biomass accumulation, particularly in regions that have historically faced water stress. This observation aligns with the findings of Zhang et al. [66], who emphasized the crucial role of precipitation in maintaining grassland productivity. By exploring the relationship between these climatic trends and AGB dynamics, our research reveals the potential impact of climate change on grassland ecosystems, with implications for biodiversity and ecosystem services.

Soil moisture (SM) and grazing intensity also play critical roles in shaping AGB dynamics. As a key indicator of plant water availability, SM directly affects evapotranspiration and the physiological processes such as photosynthesis and plant growth. When SM is higher, it enhances soil water availability to plants, which improves water uptake, promotes metabolic processes, and ultimately supports vegetation growth. This leads to an increase in AGB [59]. In contrast, low SM can reduce water availability, causing water stress that hampers plant growth and reduces AGB. Additionally, increased grazing intensity often reduces AGB by limiting vegetative cover and the ecosystem's capacity to utilize available water resources effectively [62]. This dual impact emphasizes the importance of sustainable grazing practices that balance livestock needs with ecological protection.

This study not only supports previous findings regarding the influence of climatic, environmental factors and human activities on grassland AGB but also provides new insights into the spatial variability of AGB responses across different grassland types in northern China and alpine regions. Addressing the challenges posed by climate change in these regions requires

concerted efforts to enhance ecological security and improve grassland management practices, ensuring the long-term sustainability and resilience of these critical ecosystems.

### 4.4. Uncertainty analysis in grassland AGB estimation

Although this study employs GIMMS NDVI and meteorological data to estimate grassland AGB, some limitations remain. The uncertainties associated with remote sensing data, particularly NDVI, could potentially impact on our findings. The GIMMS NDVI data may be affected by atmospheric conditions, sensor calibration, and the inherent limitations of NDVI in capturing biomass variations, which could introduce biases into biomass estimations [67, 68]. Moreover, due to its relatively low spatial resolutions, NDVI may not effectively capture the details of vegetation growth, leading to uncertainties in our research.

Furthermore, we utilized data from 99 grassland meteorological stations to estimate AGB through the CASA model. Although this model has been shown to be effective, integrating the varying local environmental conditions from the meteorological stations can introduce additional uncertainties in assessing AGB dynamics. The use of linear interpolation and Savitzky-Golay (SG) filters to address gaps and outliers in the GIMMS NDVI data can also introduce uncertainties [39]. Specifically, linear interpolation can oversimplify temporal variations, potentially masking real fluctuations in vegetation growth, while SG filters, although helpful in smoothing data, can alter the original signal, potentially leading to inaccuracies in biomass estimations. Together, these factors present significant challenges to accurately evaluating the AGB dynamics in grassland ecosystems.

Variations in grassland structure, density, and coverage can also influence NDVI, which complicates the interpretation of biomass data [57, 62]. Therefore, future studies should focus on integrating ground-based observations and other remote sensing techniques to improve the accuracy of AGB estimates and explore the implication for biodiversity and ecosystem services.

## 5. Conclusions

This study utilized meteorological and GIMMS NDVI remote sensing data to estimate the aboveground biomass (AGB) of grassland in northern China and alpine regions from 1981 to 2015, using the CASA model and Gill method. The analysis focused on monitoring dynamic changes and assessing the impacts of climate change on grassland productivity. Key findings include:

1. The mean annual AGB for northern China's grasslands and alpine regions was 272.72 $g/m^2$ over the study period.

2. A noticeable overall increase in AGB was observed, with a mean rate of 1.31 $g/m^2$ per year. Significant increases in biomass were observed in 39.96% of the grassland area, primarily in northwest Tibet, western Qinghai, and northwest Sinkiang.

3. The most primary driving factors influencing the spatiotemporal dynamics of AGB included annual mean precipitation, soil moisture, and grazing intensity. Precipitation and soil moisture are key drivers of grassland AGB in growing season (spring, summer, and autumn), while temperature plays a critical role in maintaining biomass accumulation in winter

The findings provide valuable insights into the impacts of climate factors and human activities on grassland ecosystems. They underscore the importance of integrating both climatic and

human-induced factors into grassland management strategies to ensure the long-term sustainability and resilience of these ecosystems.

## Supporting information

**S1 Table. AGB statistics for various grassland types in northern China and alpine grassland.**
(DOCX)

**S2 Table. The regression equations for grassland AGB across different grassland types from 1981 to 2015.**
(DOCX)

**S3 Table. The grassland AGB dynamics trends in northern China and alpine grasslands.**
(DOCX)

**S4 Table. The coefficient of variation (CV) of AGB in northern China and alpine grasslands.**
(DOCX)

**S5 Table. Correlation coefficients between AGB and temperature for different grassland types in different seasons.**
(DOCX)

**S6 Table. Correlation coefficients between AGB and precipitation for different grassland types in different seasons.**
(DOCX)

**S7 Table. Correlation coefficients between AGB and soil moisture for different grassland types in different seasons.**
(DOCX)

**S1 Fig. Temporal trends of (a) temperature and (b) precipitation in northern China and alpine grasslands from 1981 to 2015.**
(TIF)

**S2 Fig. Spatial dynamics of (a) annual mean temperature, (b) slope of temperature, (c) annual mean precipitation and (d) slope of precipitation in northern China and alpine grasslands from 1981 to 2015.** Source: Land use and cover data are from the Land Processes Distributed Active Archive Center (LP DAAC, https://lpdaac.usgs.gov).The study areas are from National Geomatics Center of China (https://www.ngcc.cn).
(TIF)

## Acknowledgments

We would like to thank the China Meteorological Administration (CMA) for providing meteorological data, Plant Data Center of Chinese Academy of Sciences (https://www.plantplus.cn) for the vegetation map of China, the National Aeronautics and Space Administration (NASA) Goddard Space Center for the GIMMS NDVI3g data, and the European Centre for Medium-Range Weather Forecasts (ECMWF)for ERA5 data. The authors are also grateful to the anonymous reviewers whose insights and suggestions have greatly enhanced this manuscript.

## Author Contributions

**Funding acquisition:** Xinyun Wang.

**Investigation:** Ji Zhu.

**Methodology:** Xinyun Wang, Ji Zhu, Peipei Pan.

**Validation:** Ji Zhu.

**Writing – original draft:** Xinyun Wang.

**Writing – review & editing:** Peipei Pan.

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
