## [Decision Letter · Decision Letter 0]

18 Oct 2024

PONE-D-24-27143Assessing dynamics of aboveground biomass in northern China's grassland in response to climate change, 1981-2015PLOS ONE

Dear Dr. Wang,

Thank you for submitting your manuscript to PLOS ONE. After careful consideration, we feel that it has merit but does not fully meet PLOS ONE’s publication criteria as it currently stands. Therefore, we invite you to submit a revised version of the manuscript that addresses the points raised during the review process.

We look forward to receiving your revised manuscript.

Kind regards,

Dafeng Hui, Ph.D.

Academic Editor

PLOS ONE

**Journal Requirements:**

This research was supported by the Natural Science Foundation of Ningxia Province (Project No. 2022AAC03097). Appreciation is extended to the China Meteorological Administration (CMA) for providing meteorological data and to National Aeronautics and Space Administration (NASA) Goddard Space Center for the GIMMS NDVI3g data. The authors are grateful to the anonymous reviewers whose insights and suggestions significantly enriched this manuscript.

**Additional Editor Comments:**

Please revise the manuscript based on the reviewer's comments.

Reviewers' comments:

Reviewer's Responses to Questions

**Comments to the Author**

1. Is the manuscript technically sound, and do the data support the conclusions?

Reviewer #1: Yes

2. Has the statistical analysis been performed appropriately and rigorously? 

Reviewer #1: Yes

3. Have the authors made all data underlying the findings in their manuscript fully available?

Reviewer #1: Yes

4. Is the manuscript presented in an intelligible fashion and written in standard English?

Reviewer #1: Yes

5. Review Comments to the Author

**Reviewer #1:** Based on the NDVI data and meteorological data, this study estimated the AGB in northern China’s temperate and alpine grassland and its relationship with climatic variables. The results of this study may contribute to further understanding the relationships between AGB and climate change. However, there are some concerns that the authors should address before it can be accepted for publication.

1. In the introduction, I suggest the authors add some contents about how environmental drivers influence the dynamics of grassland AGB.

2. In terms of the part two-materials and methods, it is better to simplify and integrate the part of data source and data analysis, as the classification may be inaccurate and the contents are a little prolix.

3. Since data from 99 meteorological stations are used, it is better to add the distribution of meteorological stations.

4. I suggest the authors add some contents on handling missing values and outliers in NDVI data and meteorological data.

5. In order to provide a comprehensive assessment of grassland AGB in Northern China, I suggest the authors add some time-scale analysis of grassland AGB.

6. When drawing pictures, pay attention to color brightness and saturation to give people a sense of comfort, the authors should modify the figure 5 and 6.

7. More mechanism explanations should be added to further explain the influence of driving factors on the dynamics of grassland vegetation.

8. In order to further highlight the innovation of this article, it is better to compare the results of this study with some related studies.

9. In the uncertainty, I suggest the authors further discuss the uncertainty of remote sensing data including NDVI data (e.g., Shen et al., 2021; Wang et al., 2021) which may affect the research results.

References:

Aboveground biomass and its spatial distribution pattern of herbaceous marsh vegetation in China. Science China Earth Sciences, 2021, 64: 1115-1125.

Spatiotemporal change of aboveground biomass and its response to climate change in marshes of the Tibetan Plateau. International Journal of Applied Earth Observation and Geoinformation, 2021, 102: 102385.

6. PLOS authors have the option to publish the peer review history of their article (what does this mean?). If published, this will include your full peer review and any attached files.

Reviewer #1: No

---

## [Author Response · Author response to Decision Letter 0]

19 Nov 2024

Journal Requirements:

1. When submitting your revision, we need you to address these addition requirements.

Response: We have modified the manuscript format, including the title, author information, affiliations, and overall manuscript structure, in accordance with PLOS ONE submission guidelines. 

Response: We have updated the funding statement to reflect the support we received, as per the grant information.

3. Thank you for stating the following in the Acknowledgments Section of your manuscript: funding information should not appear in the Acknowledgments section or other areas of your manuscript.

Response: AS requested, we have removed all funding-related text from the acknowledgments section, in line with PLOS ONE's guidelines.

4. When completing the data availability statement of the submission form, you indicated that you will make your data available on acceptance.

Response: We confirm that we will make our data available upon acceptance of the manuscript. The data will be shared in accordance with the journal's data share policy.

5. Please include captions for your Supporting Information files at the end of your manuscript, and update any in-text citations to match accordingly.

Response: We have added the captions for all Supporting Information files at the end of the revised manuscript. We have also updated the in-text citations to ensure they correctly correspond to these files (e.g., S1 Fig.tif and S2 Fig.tif). 

6. Please review your reference list to ensure that it is complete and correct.

Response: We have reviewed and updated the reference list, ensuring it is complete and formatted correctly. We have added new references (highlighted in yellow), and removed some older references as per the reviewer's.

Comments to the Author

1. Is the manuscript technically sound, and do the data support the conclusions?

Response: We have ensured that all remote sensing, meteorological and ground-measured data sources from public data and processing methods have commonly been utilized. The conclusions are drawn based on the data presented.

2. Has the statistical analysis been performed appropriately and rigorously?

Response: We have checked the statistical analysis to ensure it can be executed correctly according to the methods introduced in the revised manuscript.

3. Have the authors made all data underlying the findings in their manuscript fully available?

Response: Yes. All data underlying the findings can be found from 10.5281/zenodo.14038582.

4. Is the manuscript presented in an intelligible fashion and written in standard English?

Response: We have carefully reviewed the manuscript written in standard English and ensure that it is grammatically correct. We have corrected any minor printing errors. 

5. Review Comments to the Author

According to PLOS ONE data policy, the data underlying the findings have been made available through 10.5281/zenodo.14038582. The data analysis methods were performed using standard methods to ensure data right. This manuscript is not under consideration for publication elsewhere, and there are no concerns regarding duplicate publication. 

Reviewer #1: Based on the NDVI data and meteorological data, this study estimated the AGB in northern China’s temperate and alpine grassland and its relationship with climatic variables. The results of this study may contribute to further understanding the relationships between AGB and climate change. However, there are some concerns that the authors should address before it can be accepted for publication.

1. In the introduction, I suggest the authors add some contents about how environmental drivers influence the dynamics of grassland AGB.

Response: We appreciate your suggestion to expand on how environmental drivers influence the dynamics of aboveground biomass (AGB). We have added some new contents in the induction section that explains the role of climatic factors, environmental conditions, and human activities in shaping AGB dynamics. Additionally, we detailed on the impacts of soil properties, nutrient availability, and soil moisture on vegetation growth and biomass accumulation, as well as human activities like grazing intensity can alter the structure and composition of plant communities, ultimately impacting AGB. We also explain the significance of studying northern grassland AGB and the impacts of driving factors on grassland AGB dynamics. 

2. In terms of the part two-materials and methods, it is better to simplify and integrate the part of data source and data analysis, as the classification may be inaccurate and the contents are a little prolix.

Response: We have revised the “Materials and Methods” section to simplify and integrate the data source and data analysis parts, aiming for clearer organization and conciseness. This should enhance the readability and logical flow of the manuscript. 

3. Since data from 99 meteorological stations are used, it is better to add the distribution of meteorological stations.

Response: Thank you for your valuable suggestion. We have taken the reviewer’s suggestion into consideration and have added the distribution of climate stations in Fig 1 as requested. 

4. I suggest the authors add some contents on handling missing values and outliers in NDVI data and meteorological data.

Response: We have added a detailed explanation of how we handled missing values and outliers in both the GIMMS NDVI and meteorological data in the manuscript. Missing values in NDVI data were addressed using linear interpolation. Savitzky-Golay filters was used to smooth gaps. Outlier in NDVI data were detected and removed using the interquartile range (IQR) method. For the meteorological data, we employed linear interpolation to handle missing values. Outlier were detected and addressed using the IQR method to ensure data integrity. We believe that these detailed explanations enhance the clarity of our data preprocessing procedure. 

5. In order to provide a comprehensive assessment of grassland AGB in Northern China, I suggest the authors add some time-scale analysis of grassland AGB.

Response: We have added a new section analyzing the seasonal changes in grassland AGB. We have revised the structure: 3.2 Spatial distributions of annual and seasonal changes of grassland AGB; 3.2.1 Annual distribution of grassland AGB; 3.2.2 Seasonal distribution of grassland AGB. In new section, we analyzed the changes of grassland AGB across four seasons: spring, summer, autumn, and winter. We analyzed the various seasonal fluctuations, which may be influenced by variations in precipitation and temperature across season.

6. When drawing pictures, pay attention to color brightness and saturation to give people a sense of comfort, the authors should modify the figure 5 and 6.

Response: In the revised manuscript, figures 5 and 6 have been updated and are now labeled as Figs 6 and 7. These figures were created using ArcMap software. We have adjusted the color brightness and saturation to improve visual comfort. 

7. More mechanism explanations should be added to further explain the influence of driving factors on the dynamics of grassland vegetation.

Response: We have made some revisions to address the suggestion for more mechanism explanations. We have added a new figure (Figs 7f and 7g) showing the relationship between AGB and soil moisture and grazing intensity. We relocated origin Fig 6f to Fig 8 for improved clarity. 

In the revised manuscript, we provide a comprehensive analysis of the mechanism through which climate factors, such as temperature (T), precipitation (PPT), VPD, relative humidity (RH), and solar radiation (SR), as well as human activities like grazing intensity impact on grassland AGB. 

In Fig 8, we discuss the important roles that these climate factors and human activities play in promoting plant growth and biomass accumulation, thereby providing a deeper understanding of the dynamics of grassland. 

In Fig 9, we added figure 9f to analyze the relationship between AGB and grazing intensity (GI). This addition allowed us to explore the mechanisms by which climate factors and grazing intensity impact grassland AGB, providing deeper insights into their interactions and effects on biomass dynamics.

These additions and adjustments enhance the mechanism explanation of how different factors affect grassland biomass.

8. In order to further highlight the innovation of this article, it is better to compare the results of this study with some related studies.

Response: We have revised the discussion section to include comparisons with related studies and to elaborate on the mechanism revealed by our research. We highlighted the responses of different grassland types to climate factors and human activities. This revision provides a clearer understanding of grassland dynamics and the complexities involved in managing these grassland resources effectively. 

9. In the uncertainty, I suggest the authors further discuss the uncertainty of remote sensing data including NDVI data (e.g., Shen et al., 2021; Wang et al., 2021) which may affect the research results.

References:

Aboveground biomass and its spatial distribution pattern of herbaceous marsh vegetation in China. Science China Earth Sciences, 2021, 64: 1115-1125.

Spatiotemporal change of aboveground biomass and its response to climate change in marshes of the Tibetan Plateau. International Journal of Applied Earth Observation and Geoinformation, 2021, 102: 102385.

Response: In the revised manuscript, we have included a new discussion section titled “4.4 Assessing uncertainty in grassland AGB estimation.” In this section, we analyzed the uncertainties associated with NDVI and meteorological data, as well as the potential impacts of data preprocessing on our findings. This addition provides a clearer understanding of how these uncertainties affect the accuracy of grassland AGB estimations. 

In the new submission, we have utilized Preflight Analysis and Conversion Engine (PACE) digital diagnostic tool to our figure and table files and ensure that figures and tables meet PLOS ONE requirements.

---

## [Editor Report · Decision Letter 1]

25 Nov 2024

Spatiotemporal dynamics of grassland aboveground biomass in northern China and the alpine region: Impacts of climate change and human activities

PONE-D-24-27143R1

Dear Dr. Wang,

We’re pleased to inform you that your manuscript has been judged scientifically suitable for publication and will be formally accepted for publication once it meets all outstanding technical requirements.

Kind regards,

Dafeng Hui, Ph.D.

Academic Editor

PLOS ONE

Additional Editor Comments (optional):

The authors have adequately addressed the reviewer's concerns.
---

## [Editor Report · Acceptance letter]

5 Dec 2024

PONE-D-24-27143R1 

PLOS ONE

Dear Dr. Wang, 

I'm pleased to inform you that your manuscript has been deemed suitable for publication in PLOS ONE. Congratulations! Your manuscript is now being handed over to our production team.

Kind regards, 

on behalf of

Dr. Dafeng Hui 

Academic Editor

PLOS ONE